# ABNet: Attention BarrierNet for Safe and Scalable Robot Learning

## Abstract

Safe learning is central to AI-enabled robots where a single failure may lead to catastrophic results. Barrier-based method is one of the dominant approaches for safe robot learning. However, this method is not scalable, hard to train, and tends to generate unstable signals under noisy inputs that are challenging to be deployed for robots. To address these challenges, we propose a novel Attention BarrierNet (ABNet) that is scalable to build larger foundational safe models in an incremental manner. Each head of BarrierNet in the ABNet could learn safe robot control policies from different features and focus on specific part of the observation. In this way, we do not need to one-shotly construct a large model for complex tasks, which significantly facilitates the training of the model while ensuring its stable output. Most importantly, we can still formally prove the safety guarantees of the ABNet. We demonstrate the strength of ABNet in 2D robot obstacle avoidance, safe robot manipulation, and vision-based end-to-end autonomous driving, with results showing much better robustness and guarantees over existing models.

## 1 Introduction

A Safety-Guaranteed Learning System with Attention Mechanism

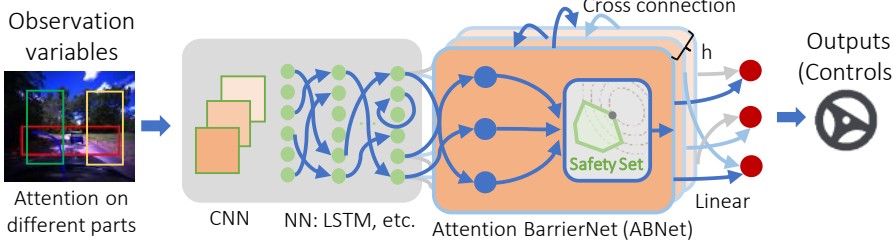

Figure 1: The proposed ABNet that is robust, scalable and generates stable output while guaranteeing safety for robots. Each head of BarrierNet in the model could learn safe control policies with attention on different observation feature in a scalable or one-shot/direct manner.

Robot learning usually requires to leverage scalable training and vast amount of data. There are many large models Li et al. (2022) for complex robotic tasks including manipulation, locomotion, autonomous driving Bommasani et al. (2021) Singh et al. (2023) Wang et al. (2023a). However, these models are not trustworthy and have no safety guarantees. Existing methods that incorporate guarantees or certificates into neural networks are not scalable and hard to train Pereira et al. (2020) Xiao et al. (2023a) Wang et al. (2023b). It is desirable to merge these models as we can get better performance controllers in general Beygelzimer et al. (2015) Agarwal et al. (2020). Traditional mixture of expert methods Shazeer et al. (2017) Riquelme et al. (2021) Zhou et al. (2022) or other merging approaches Huang et al. (2023) Ramé et al. (2023) Wang et al. (2024) are hard to retain the safety of the models. In this work, we explore to leverage the collective power of many safety-critical models to handle complex tasks while preserving the safety of the models.

There are various definitions of safety for robotics and autonomy, and safety can be basically defined as something bad never happens. Mathematically, safety can be defined as a continuously

differentiable constraint with respect to the system state and it can be further captured by the forward invariance of the safe set over such a constraint Ames et al. (2017) Xiao & Belta (2022) Glotfelter et al. (2017). In other words, we can use different constraints and approaches to enforce safety. The way we learn such safety enforcement methods may depend on the focused observation feature, which corresponds to the attention mechanism. For instance, some human drivers may focus on the left lane boundary in driving in order to achieve safe lane keeping, while others may focus on the right lane boundary, as shown in Fig. 1. Both attention mechanisms can achieve similar purpose. Merging these models or attention mechanisms enables us to build robust and powerful learning models. However, retaining safety is non-trivial.

In the literature, barrier-based learning methods Robey et al. (2020) Pereira et al. (2020) Srinivasan et al. (2020) Xiao et al. (2023b), such as the BarrierNet Xiao et al. (2023a) Wang et al. (2023b) Liu et al. (2023), are widely used to equip deep learning systems with safety guarantees. We may incorporate control-theoretic based optimizations into learning systems in the form of differentiable quadratic programs (dQPs) Amos & Kolter (2017). There are several limitations of these barrier-based learning methods: (*i*) it can only implement a single safety enforcement method as the last layer of the neural network, which is not scalable to larger safe learning models; (*ii*) the model is not robust such that it is hard to be trained to work for complicated robotic applications; (*iii*) these methods tend to generate unstable output under noisy observation, which is intractable to be deployed for robots.

In this paper, we propose a novel Attention BarrierNet (ABNet) to merge many safety-critical models while preserving the safety guarantees.The ABNet is scalable, robust to noise, and easy to be trained in an incremental manner. As shown in Fig. 1, we may build multi-head BarrierNets within the ABNet. Each head of the BarrierNet may pay attention to different observation features to generate a safe control policy. We linearly combine the outputs of all the BarrierNets in a way that is provably safe. The weights of this combination quantify the importance of each head of BarrierNet, and they are trainable. The structure of the ABNet allows us to build larger foundational safe models for various and complicated robotic applications as we can incrementally train safe models corresponding to different robot skills and this will simply increase the head $h$ of BarrierNets.

In summary, we make the following **new contributions**:

- We propose a novel ABNet that merges many safety-critical learning models, and this new model is scalable, robust, and easy to be trained.
- We formally prove the safety guarantees of the proposed ABNet.
- We demonstrate the strength and effectiveness of our model on a variety of robot control tasks, including 2D robot obstacle avoidance, safe robot manipulation, and vision-based end-to-end autonomous driving in an open dataset. We also show that existing models/policies merging could make safety worse in complicated tasks (such as in vision-based driving).

## 2 PRELIMINARIES AND PROBLEM FORMULATION

In this section, we present background on the forward invariance with High-Order Control Barrier Functions (HOCBFs) that is widely used to enforce safety, as well as introduce the BarrierNet.

**Forward Invariance with HOCBFs.** Consider an affine control system defined as:

$$\dot{x} = f(x) + g(x)u \tag{1}$$

where $x \in \mathbb{R}^n$ is the system state, $f : \mathbb{R}^n \to \mathbb{R}^n$ and $g : \mathbb{R}^n \to \mathbb{R}^{n \times q}$ are locally Lipschitz, and $u \in U \subset \mathbb{R}^q$, where $U$ denotes a control constraint set. $\dot{x}$ denotes the time derivative of state $x$. We can also consider non-affine control systems by defining auxiliary systems Xiao et al. (2023b).

Consider a safety constraint $b(x) \geq 0$ with relative degree $m$ (i.e., we need to differentiate $b(x)$ $m$ times along the dynamics (1) until any controls first show up in the derivative) for system (1), where $b : \mathbb{R}^n \to \mathbb{R}$ is continuously differentiable, we recursively define a sequence of CBFs $\psi_i : \mathbb{R}^n \to \mathbb{R}, i \in \{1, \ldots, m\}$ in the form ($\psi_0(x) := b(x)$):

$$\psi_i(x) := \dot{\psi}_{i-1}(x) + \alpha_i(\psi_{i-1}(x)), i \in \{1, \ldots, m\}, \tag{2}$$

where $\alpha_i, i \in \{1, \ldots, m\}$ are class $\mathscr{K}$ functions (strictly increasing that passes through the origin).

We further define a sequence of safe sets $C_i, i \in \{1 \ldots, m\}$ corresponding to (2) in the form:

$$C_i := \{x \in \mathbb{R}^n : \psi_{i-1}(x) \geq 0\}, i \in \{1, \ldots, m\}. \tag{3}$$

**Definition 2.1.** *(High Order Control Barrier Function (HOCBF) Xiao & Belta (2022)): Let $C_i, i \in \{1, \ldots, m\}$ and $\psi_i, i \in \{1, \ldots, m\}$ be defined by (3) and (2), respectively. A function $b : \mathbb{R}^n \to \mathbb{R}$ is a HOCBF if there exist class $\mathscr{K}$ functions $\alpha_i, i \in \{1 \ldots, m\}$ such that*

$$\sup_{\boldsymbol{u} \in U} [L_f \psi_{m-1}(\boldsymbol{x}) + [L_g \psi_{m-1}(\boldsymbol{x})] \boldsymbol{u} + \alpha_m(\psi_{m-1}(\boldsymbol{x}))] \geq 0, \tag{4}$$

*for all $\boldsymbol{x} \in \cap_{i=1}^m C_i$. $L_f \psi_{m-1}(\boldsymbol{x}) = \frac{d\psi_{m-1}(\boldsymbol{x})}{d\boldsymbol{x}} f(x)$ and $L_g \psi_{m-1}(\boldsymbol{x}) = \frac{d\psi_{m-1}(\boldsymbol{x})}{d\boldsymbol{x}} g(x)$.*

The following theorem shows the safety guarantees of HOCBFs:

**Theorem 2.2** (Xiao & Belta (2022))**.** *Given a HOCBF $b(\boldsymbol{x})$ from Def. 2.1, if $\boldsymbol{x}(0) \in \cap_{i=1}^m C_i$, then any Lipschitz continuous controller $\boldsymbol{u}(t)$ that satisfies the constraint in (4), $\forall t \geq 0$ renders $\cap_{i=1}^m C_i$ forward invariant for system (1), i.e., $b(\boldsymbol{x}(t)) \geq 0, \forall t \geq 0$.*

The HOCBF is required for high-relative-degree systems, and it is a general form of the CBF Ames et al. (2017) Glotfelter et al. (2017), i.e., setting the relative degree $m = 1$ of a safety constraint $b(\boldsymbol{x}) \geq 0$ will reduce a HOCBF to a CBF. CBFs/HOCBFs are widely used to transform nonlinear optimal control problems into a sequence of Quadratic Programs (QPs) that are very efficient to solve while preserving the safety guarantees of the system.

**BarrierNet.** The BarrierNet Xiao et al. (2023a) is a neural network layer that incorporates CBF/HOCBF-based QPs as differentiable QPs (dQPs) Amos & Kolter (2017), in which all the CBFs/HOCBFs are differentiable in terms of their parameters (such as those in class $\mathscr{K}$ functions). Those parameters are crucial to the system conservativeness or performance in guaranteeing safety. In summary, the BarrierNet frees us from handing-tuning all the parameters in safety-critical controls, and simply uses data to optimize them. It can be trained using either imitation learning Xiao et al. (2023a) or reinforcement learning Liu et al. (2023). Referring to Fig. 1, a BarrierNet only has a single head in the model (i.e., $h = 1$) and it is placed as the last layer of the model when used in conjunction with other neural networks (such as CNN and LSTM).

In this paper, we consider the following safe learning problem:

**Problem.** Given (a) a system with dynamics in the form of (1); (b) a state-feedback nominal controller $\pi^*(\boldsymbol{x}) = \boldsymbol{u}^*$ (such as a model predictive controller) that provides the training label; (c) a set of safety constraints $b_j(\boldsymbol{x}) \geq 0, j \in S$ ($b_j$ is continuously differentiable, $S$ is a constraint set); (d) a neural network controller $\pi(\boldsymbol{x}, \boldsymbol{z} | \boldsymbol{\theta}) = \boldsymbol{u}$ parameterized by $\boldsymbol{\theta}$ (under observation $\boldsymbol{z}$);

Our goal is to find the optimal parameter

$$\boldsymbol{\theta}^* = \arg \min_{\boldsymbol{\theta}} \mathbb{E}_{\boldsymbol{x}, \boldsymbol{z}} [\ell(\pi^*(\boldsymbol{x}), \pi(\boldsymbol{x}, \boldsymbol{z} | \boldsymbol{\theta}))], \tag{5}$$

while satisfying all the safety constraints in (c) and the dynamics constraint (a). $\mathbb{E}$ is the expectation, and $\ell$ is a loss function.

## 3 ATTENTION BARRIERNET

In this section, we present the architecture of the Attention BarrierNet (ABNet) and formally prove its safety guarantees in learning systems.

### 3.1 MULTI-HEAD BARRIERNETS

We can use a BarrierNet to transform the constrained optimal control in the considered problem into the following differentiable QP, which forms a head of BarrierNet in the model:

$$\boldsymbol{u}_k = \arg \min_{\boldsymbol{u}(t) \in U} \frac{1}{2} \boldsymbol{u}(t)^T H(\boldsymbol{z}_k | \boldsymbol{\theta}_{h,k}) \boldsymbol{u}(t) + F^T(\boldsymbol{z}_k | \boldsymbol{\theta}_{f,k}) \boldsymbol{u}(t) \tag{6}$$

s.t.

$$L_f \psi_{j,m-1}(\boldsymbol{x}, \boldsymbol{z} | \boldsymbol{\theta}_p) + [L_g \psi_{j,m-1}(\boldsymbol{x}, \boldsymbol{z} | \boldsymbol{\theta}_p)] \boldsymbol{u} + p_{m,k}(\boldsymbol{z}_k | \boldsymbol{\theta}_{p,k}^m) \alpha_{j,m}(\psi_{j,m-1}(\boldsymbol{x}, \boldsymbol{z} | \boldsymbol{\theta}_p)) \geq 0, j \in S,$$

$$\psi_{j,i}(\boldsymbol{x}, \boldsymbol{z} | \boldsymbol{\theta}_p) = \dot{\psi}_{j,i-1}(\boldsymbol{x}, \boldsymbol{z} | \boldsymbol{\theta}_p) + p_i(\boldsymbol{z} | \boldsymbol{\theta}_p^i) \alpha_{j,i}(\psi_{j,i-1}(\boldsymbol{x}, \boldsymbol{z} | \boldsymbol{\theta}_p)), i \in \{1, \ldots, m-1\}, j \in S, \tag{7}$$

$$\psi_{j,0}(\boldsymbol{x}, \boldsymbol{z} | \boldsymbol{\theta}_p) = b_j(\boldsymbol{x}), j \in S, \qquad t = \omega \Delta t + t_0, \omega \in \{0, 1, \ldots\},$$

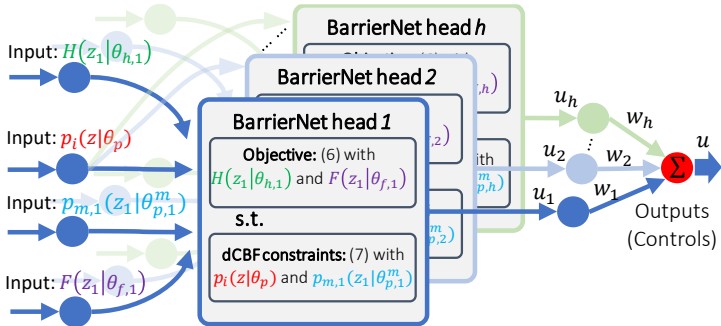

Figure 2: Architecture of multi-head BarrierNets (i.e., ABNet). The ABNet is usually used in conjunction with any other neural networks and can be implemented in parallel. The parameters (inputs) of each head of BarrierNet are the outputs of previous layers (such as CNN or LSTM).

where $k \in \{1, \ldots, h\}$, and $h$ is the number of heads of BarrierNet (as shown in Fig. 1). $p_i \geq 0, i \in \{1, \ldots, m-1\}, p_{m,k} \geq 0$ are penalty functions on the class $\mathscr{K}$ functions $\alpha_{j,i}, i \in \{1, \ldots, m\}, j \in S$ that address the conservativeness of the model (e.g., how far away the system state should stay form the unsafe set bound in order to maintain safety). All the HOCBFs corresponding to the safety constraints share the same penalty functions, but they may use different ones in which case $p_i$ and $p_{m,k}$ will be dependent on $j, j \in S$. The derivatives of the observation $z$ in the above are omitted, as shown in Xiao et al. (2023a). $H(z_k|\theta_{h,k}) \in \mathbb{R}^{q \times q}$ is positive definite, and $H^{-1}(z_k|\theta_{h,k})F(z_k|\theta_{f,k})$ can be interpreted as a reference control (the output of previous network layers). $\theta := (\theta_h, k, \theta_{f,k}, \theta_{p,k}^m, \theta_p), k \in \{1, \ldots, h\}$, where $\theta_p := (\theta_p^1, \ldots, \theta_p^{m-1})$ are all trainable parameters of the neural network. $z_k$ is the observation of the BarrierNet head $k, k \in \{1, \ldots, h\}$, and it is possible that all heads share the same observation, i.e. $z_k = z, \forall k \in \{1, \ldots, h\}$. $\Delta t > 0$ is the discretized time interval, and $t_0$ is the initial time.

**Attention mechanism.** Each head of BarrierNet may learn safe self-attention even if all the BarrierNets have the same observation $z$. The parameter $p_{m,k}^m$ may be learned from different input features via random initialization, and it determines the conservativeness of the model in guaranteeing safety. On the other hand, we may also make each head of BarrierNet focus on different observations $z_k$. The observation $z_k$ may come from different parts of the sensor observation (such as the left lane boundary and right lane boundary in driving shown in Fig. 1), or even different perceptions (such as vision, lidar, etc.)

**Cross connection.** It can be noted from (7) that each head of BarrierNet $k \in \{1, \ldots, h\}$ has some cross connection with other heads, as also shown in Fig. 1. In other words, $\psi_{j,i}(x, z|\theta_p), i \in \{1, \ldots, m-1\}, j \in S$ are formulated in the same way through the shared parameter $\theta_p$ (independent from $k$). This is to ensure (i) the construction for provable safety (as shown later), and (ii) some shared information across different heads of BarrierNet as they all generate safe controls for (1).

**Fusion.** Another important consideration is how should we fuse all these controls $u_k, k \in \{1, \ldots, h\}$ while preserving the safety property of each head of the BarrierNet. We propose the following form:

$$u = \sum_{k=1}^{h} w_k u_k, \quad \text{where} \sum_{k=1}^{h} w_k = 1. \tag{8}$$

In the above, $w_k \geq 0, k \in \{1, \ldots, h\}$ are trainable parameters. The composition of all the heads of BarrierNet (6) s.t., (7) in the form of (8) is our proposed *ABNet*, as shown in Fig. 2. The safety guarantees of the ABNet is shown in the following theorem:

**Theorem 3.1.** *(Safety of ABNets) Given the multi-head BarrierNets formulated as in (6) s.t. (7). If the system (1) is initially safe (i.e., $b_j(x(t_0)) \geq 0, \forall j \in S$), then a control policy $u$ from the ABNet output (8) guarantees the safety of system (1), i.e., $b_j(x(t)) \geq 0, \forall j \in S, \forall t \geq t_0$.*

All the proofs for theorems are given in Appendix A. If the system is not initially safe (i.e., $b_j(x(t_0)) < 0, \exists j \in S$), then the system state $x$ of (1) will be driven to the safe side of the state space due to the Lyapunov property of CBF/HOCBFs Ames et al. (2017) Xiao & Belta (2022). This enables the possibility of utilizing data that violates safety to conduct adversary training of the ABNet.

---

**Algorithm 1** Construction and training of ABNet

---

**Input:** the problem setup (a)-(d) given in the problem formulation (end of Sec. 2).
**Output:** a robust and safe controller $\boldsymbol{u}$ for system (1).
(a) Formulate each head of BarrierNet as in (6) s.t. (7).
(b) Build the cross connection among BarrierNets via $p_i(\boldsymbol{z}|\boldsymbol{\theta}_p^i), i \in \{1,\ldots,m-1\}$.
(c) Fuse all the heads of BarrierNet as in (8).
**if** *Scalable training* **then**
    Decouple $p_i(\boldsymbol{z}|\boldsymbol{\theta}_p^i), i \in \{1,\ldots,m-1\}$ and define them for each BarrierNet.
    Train each head of BarrierNet, respectively.
    Choose a $p_i(\boldsymbol{z}|\boldsymbol{\theta}_p^i), i \in \{1,\ldots,m-1\}$ from one of the BarrierNets to build cross connection.
    Fuse all the BarrierNets via (9).
**else**
    Directly train the ABNet via reverse mode error back propagation.
**end if**

---

**Natural noise filter.** The ABNet is a natural noise filter since $w_k \in [0,1], \forall k \in \{1,\ldots,h\}$ in (8). This can ensure that the output $\boldsymbol{u}$ of the model is stable with a large enough head number $h$ if all the BarrierNets have different observation $\boldsymbol{z}_k$ for the current environment. This feature makes ABNet a very robust controller for robotic systems, and thus, ABNet can generate smooth signals.

**Theorem 3.2.** *(Safety of merging of ABNets) Given two ABNets with each formulated as in (8) and (6) s.t. (7), the merged model using the form as in (8) again guarantees the safety of system (1).*

## 3.2 MODEL TRAINING

The ABNet can be trained incrementally or in one-shot. This is due to the fact that each head of BarrierNet can generate a control policy that is applicable to system (1). The linear combination weights $w_k, k \in \{1,\ldots,h\}$ in the ABNet denote the importance of the corresponding control policies.

**Scalable training.** In ABNet, we may train each head $k, k \in \{1,\ldots,h\}$ of the BarrierNet in a scalable way as we wish to minimize the loss between their output $\boldsymbol{u}_k$ and the label $\boldsymbol{u}^*$ as well. The training can be done using the batch QP training method proposed in Amos & Kolter (2017). There are some cross connections via $p_i(\boldsymbol{z}|\boldsymbol{\theta}_p)$ between BarrierNets in the ABNet that may prevent the implementation of the training. We may address this by training a $p_i(\boldsymbol{z}|\boldsymbol{\theta}_p)$ for each head of the BarrierNet. After we train all heads of the BarrierNet, we may fix the parameters of those models, choose a $p_i(\boldsymbol{z}|\boldsymbol{\theta}_p)$ from one of the BarrierNets (or take an average of all $p_i(\boldsymbol{z}|\boldsymbol{\theta}_p)$ among the BarrierNets) to build the cross connection, and train the weights $w_i$ for some more iterations. Another way is to fuse these BarrierNets by their testing loss. In other words, the weight $w_k, k \in \{1,\ldots,h\}$ can be determined by:

$$w_k = \frac{1/\ell_k(\boldsymbol{u}_k,\boldsymbol{u}^*)}{\sum_{k=1}^h 1/\ell_k(\boldsymbol{u}_k,\boldsymbol{u}^*)}, \tag{9}$$

where $\ell_k$ is a loss function.

If we already have some trained ABNet, and we wish to add some new capabilities (such as safe driving by only focusing on the left lane boundary) to the model, then we can train some heads of BarrierNets based on the new data we have. Finally, we can fuse the models similarly with safety guarantees as shown in Thm. 3.2. This shows the scalability of the proposed ABNet that allows us to build larger foundational safe models in an incremental way.

**One-shot/Direct training**. The one-shot training of the ABNet can be directly done using the traditional reverse mode automatic differentiation. In addition to the loss between the eventual output $\boldsymbol{u}$ of the ABNet and the label $\boldsymbol{u}^*$, we may also consider the losses on $\boldsymbol{u}_k, k \in \{1,\ldots,h\}$, as well as on the reference controls $H^{-1}(\boldsymbol{z}_k|\boldsymbol{\theta}_{h,k})F(\boldsymbol{z}_k|\boldsymbol{\theta}_{f,k})$, in order to improve the training performance.

The construction and training of the ABNet involve the formulation of each head of BarrierNet as in (6) s.t. (7), the BarrierNet fusion as in (8), and the scalable or direct training as shown above (Alg. 1).

**Computational efficiency.** The training and deployment efficiency is similar to that of a single head ABNet (i.e., a BarrierNet) as we can implement all the heads of a ABNet in parallel.

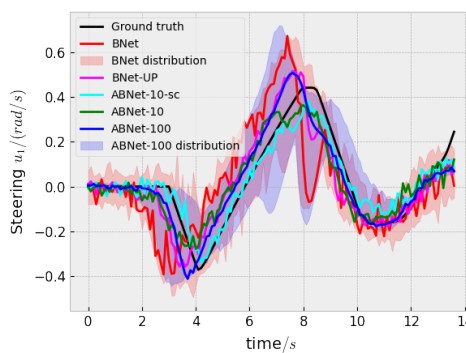 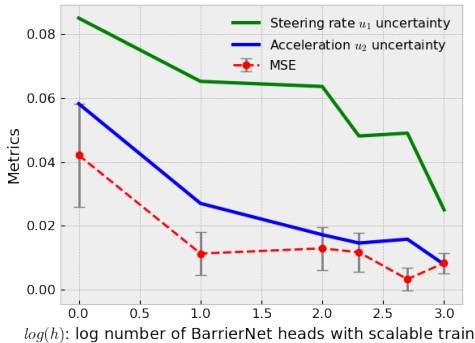

Figure 3: 2D robot obstacle avoidance closed-loop testing control profiles (left) and ABNet performance with the increasing of BarrierNet heads using scalable training (right). This scalable training for ABNet is with safety guarantees. The controls are subject to input noise, and thus are non-smooth.

## 4 EXPERIMENTS

Table 1: 2D robot obstacle avoidance closed-loop testing under noisy input.

| MODEL | SAFETY ($\geq 0$) | CONSER. ($\geq 0$ & $\downarrow$) | MSE($\downarrow$) | $u_1$ UNCER-TAINTY ($\downarrow$) | $u_2$ UNCER-TAINTY ($\downarrow$) | THEORET. GUAR. |
|---|---|---|---|---|---|---|
| E2E LEVINE ET AL. (2016) | -14.140 | $-2.976 \pm 3.770$ | $0.007 \pm 0.004$ | 0.063 | 0.049 | $\times$ |
| E2Es-MCD GAL & GHAHRAMANI (2016) | -2.087 | $-1.341 \pm 0.824$ | $0.004 \pm 0.001$ | 0.041 | 0.026 | $\times$ |
| E2Es-DR LAKSHMINARAYANAN ET AL. (2017) | -35.130 | $-3.176 \pm 4.299$ | $0.080 \pm 0.006$ | 0.032 | 0.020 | $\times$ |
| DFB PEREIRA ET AL. (2020) | 36.659 | $47.810 \pm 4.377$ | $0.013 \pm 0.003$ | 0.062 | 0.052 | $\checkmark$ |
| BNET XIAO ET AL. (2023a) | 5.045 | $7.966 \pm 1.287$ | $0.014 \pm 0.006$ | 0.074 | 0.047 | $\checkmark$ |
| BNET-UP WANG ET AL. (2023b) | 5.988 | $8.573 \pm 1.738$ | $0.008 \pm 0.004$ | 0.054 | 0.028 | $\times$ |
| ABNET-10-SC (OURS) | 5.731 | $6.269 \pm 0.319$ | $0.011 \pm 0.007$ | 0.065 | 0.027 | $\checkmark$ |
| ABNET-10 (OURS) | 12.639 | $13.887 \pm 1.323$ | $0.008 \pm 0.005$ | 0.049 | 0.030 | $\checkmark$ |
| ABNET-100 (OURS) | 10.122 | $11.729 \pm 0.816$ | $0.012 \pm 0.006$ | 0.049 | 0.013 | $\checkmark$ |

In this section, we conduct several experiments to answer the following questions:

• Does our method match the theoretic results in experiments and **is it scalable**?

• How does our method compare with state-of-the-art models in enforcing safety constraints?

• The benefit of models/policies merging and the robustness of our models in safety and smoothness?

**Benchmark models:** We compare with (i) *baseline*: **Tables 1, 2**–single end-to-end learning model (E2E) Levine et al. (2016) and **Table 3**–single vanilla end-to-end (V-E2E) model Amini et al. (2022), (ii) *safety guaranteed models*: single BarrierNet (BNet) Xiao et al. (2023a), Deep forward and backward (DFB) model Pereira et al. (2020), (iii) *policies merging*: BarrierNet policies merged with uncertainty propagation (BNet-UP) Wang et al. (2023b) that employs Gaussian kernels with Scott's rule Scott (2015) to select the bandwidth, (iv) *models merging*: E2Es merged with Monte-Carlo Dropout (E2Es-MCD) Gal & Ghahramani (2016), E2Es merged with Deep Resembles (E2Es-DR) Lakshminarayanan et al. (2017).

**Our models:** *Sec. 4.1 and 4.2*: ABNet trained in a scalable way with 10 heads (ABNET-10-SC), ABNet trained in one shot with 10 heads (ABNET-10), ABNet trained in one shot with 100 heads (ABNET-100). *Sec. 4.3*: our ABNet trained in one shot with 10 heads using the same input images (ABNET), ABNet with attention images and 10 heads (ABNET-ATT), our ABNet first trained with ABNET scaled/augmented by ABNET-ATT (20 heads, ABNET-SC).

**Evaluation metrics:** The evaluation metrics are defined as in Xiao et al. (2023b): mean square error of the model testing (MSE), satisfaction of safety constraints where non-negative values mean safety guarantees (SAFETY), system conservativeness (CONSER.), steering control $u_1$ uncertainty ($u_1$ UNCERTAINTY), acceleration control $u_2$ uncertainty ($u_2$ UNCERTAINTY), and theoretical safety guarantees (THEORET. GUAR.) respectively. The metrics are explicitly defined in Appendx B.

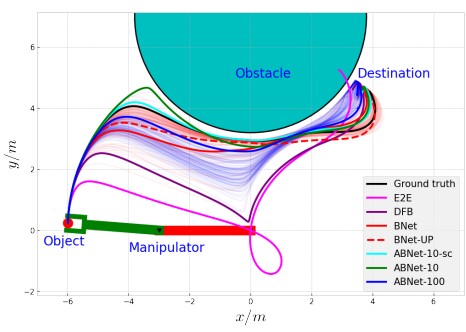 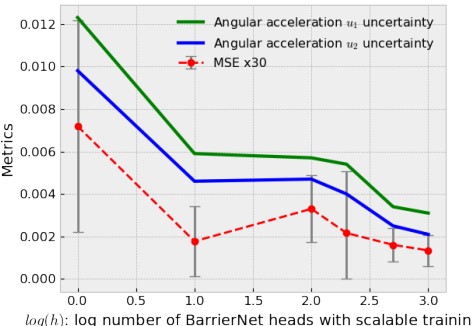

Figure 4: Robot manipulation closed-loop end-effector trajectories (left) and ABNet performance with the increasing of BarrierNet heads using scalable training (right). The transparent trajectories in the left figure are corresponding to results in all runs.

## 4.1 2D ROBOT OBSTACLE AVOIDANCE

We aim to find a neural network controller for a 2D robot that can drive the robot from an initial location to an arbitrary destination while avoiding crash onto the obstacle. All the models (h copies/heads) have the same input (with uniformly distributed noise, 10% of the input magnitude in testing). The detailed problem setup and model introductions are given in Appendix B.1. The inference times of BarrierNet, ABNet-10, ABNet-100 are similar (1.6ms) in parallel implementation (otherwise, they are 1.6ms, 14.4ms, 143.9ms, respectively)

Models/policies merging can improve the performance as shown by the MSE metrics in Table 1 and the scalable training in Fig. 3. Note that our scalable training for ABNets has safety guarantees. The DFB tends to be very conservative as the CBFs within which are not differentiable, which presents a high conservative value shown in Table 1. Our proposed ABNets can significantly reduce the uncertainty of the outputs (controls) under noisy input while guaranteeing safety, and this uncertainty decreases as the increases of the BarrierNet heads in the ABNets, as shown by the last two and three columns in Table 1, as well as shown in Fig. 3 and 6 of Appendix B.1 where the control uncertainty of ABNet-100 is lower than the one of BNet. The smoothness of the controls also increases with the increase of BarrierNet heads (e.g., blue from ABNet v.s. red from BNet in Fig. 6). In terms of performance, our proposed ABNets can also improve the testing errors compared to BNet and DFB, as shown by the MSE in Table 1. The E2Es-MCD model can achieve the best performance, but this is at the cost of safety (the SAFETY metric in Table 1 is negative, which implies violated safety).

Table 2: Robot manipulation closed-loop testing under noisy input and comparisons with benchmarks.

| MODEL | SAFETY ($\geq 0$) | CONSER. ($\geq 0$ & $\downarrow$) | MSE($\downarrow$) | $u_1$ UNCERTAINTY ($\downarrow$) | $u_2$ UNCERTAINTY ($\downarrow$) | THEORET. GUAR. |
|---|---|---|---|---|---|---|
| E2E LEVINE ET AL. (2016) | -11.027 | $-1.082\pm2.992$ | 3.6$e$-4$\pm$1.7$e$-4 | 0.013 | 0.009 | $\times$ |
| E2Es-MCD GAL & GHAHRAMANI (2016) | -11.827 | $0.162\pm2.085$ | 1.1$e$-4$\pm$7.3$e$-5 | 0.008 | 0.005 | $\times$ |
| E2Es-DR LAKSHMINARAYANAN ET AL. (2017) | -11.381 | $-0.958\pm1.875$ | 1.3$e$-4$\pm$8.5$e$-5 | 0.007 | 0.005 | $\times$ |
| DFB PEREIRA ET AL. (2020) | 2.905 | $6.023\pm3.110$ | 8.7$e$-4$\pm$1.9$e$-4 | 0.019 | 0.018 | $\checkmark$ |
| BNET XIAO ET AL. (2023A) | 0.147 | $0.745\pm0.505$ | 2.3$e$-4$\pm$1.2$e$-4 | 0.010 | 0.009 | $\checkmark$ |
| BNET-UP WANG ET AL. (2023B) | 0.206 | $0.346\pm0.098$ | 5.2$e$-5$\pm$3.2$e$-5 | 0.005 | 0.005 | $\times$ |
| ABNET-10-SC (OURS) | 0.233 | $0.570\pm0.360$ | 5.9$e$-5$\pm$5.5$e$-5 | 0.006 | 0.005 | $\checkmark$ |
| ABNET-10 (OURS) | 0.039 | $0.272\pm0.443$ | 1.2$e$-4$\pm$9.6$e$-5 | 0.008 | 0.007 | $\checkmark$ |
| ABNET-100 (OURS) | 0.053 | $0.123\pm0.177$ | 1.1$e$-4$\pm$4.4$e$-5 | 0.005 | 0.004 | $\checkmark$ |

## 4.2 SAFE ROBOT MANIPULATION

In robot manipulation, we employ a two-link planar robot manipulator to grasp an object from an arbitrary point to an arbitrary destination while avoiding crashing onto obstacles. All the models (h copies/heads) have the same input (with uniformly distributed noise, 10% of the input magnitude in testing). We compare our proposed ABNets with the same benchmark models as in the last subsection. More detailed problem setup and model introductions are given in Appendix B.2.

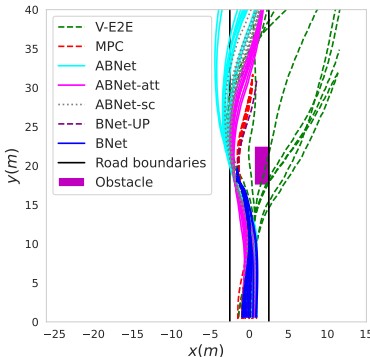 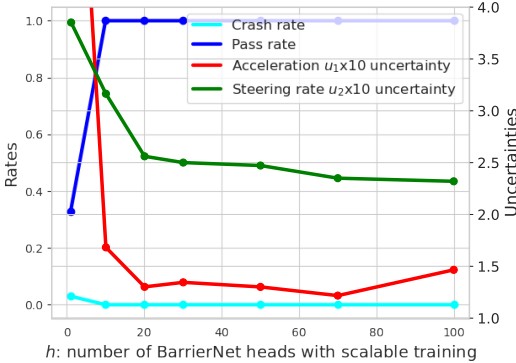

Figure 5: Vision-based end-to-end autonomous driving closed-loop testing trajectories in VISTA (left) and ABNet performance with the increasing of BarrierNet heads using scalable training (right). This scalable training is done by both the ABNet and ABNet-att in Table 3 with safety guarantees.

Again, models/policies merging can improve the performance as shown by the MSE metrics in Table 2 and the scalable training in Fig. 4. All the E2E-related models are not robust to noise and violate safety constraints (i.e., crash onto obstacles) under noisy input since there are no formal guarantees, and such an example is shown by the magenta trajectory curve of the end-effector in Fig. 4. As shown in Table 2, the proposed ABNet-100 model is the least conservative one with the lowest control uncertainties as well under noisy inputs (significantly improved compared with BNet and DFB), which demonstrates its advantage over other models. This uncertainty improvement is also shown by the control distributions in Fig. 7 in Appendix B.2 (BNet: red area v.s. ABNet-100: blue area). The BNet-UP achieves the best performance without safety guarantees.

### 4.3 VISION-BASED END-TO-END AUTONOMOUS DRIVING

We finally test our models in a more complicated and realistic task: vision-based driving, using an open dataset and benchmark from the VISTA Amini et al. (2022). One of ABNets, named ABNet-att, is constructed such that different heads of BarrierNets focus on different parts of the image (left lane boundary, right lane boundary, etc., the corresponding images are shown in Fig 8 of Appendix B.3). For more experiment and model details, please refer to Appendix B.3.

Table 3: Vision-based end-to-end autonomous driving closed-loop testing and comparisons with benchmarks. New items are short for obstacle crash rate (CRASH), obstacle passing rate (PASS).

| MODEL | CRASH ($\downarrow$) | PASS ($\uparrow$) | SAFETY ($\geq 0$) | CONSER. ($\geq 0\&\downarrow$) | $u_1$ UNCERTAINTY ($\downarrow$) | $u_2$ UNCERTAINTY ($\downarrow$) | THEORET. GUAR. |
|---|---|---|---|---|---|---|---|
| V-E2E AMINI ET AL. (2022) | 6% | 94% | -60.297 | $-0.610\pm21.165$ | 0.443 | 0.222 | $\times$ |
| E2ES-MCD GAL & GHAHRAMANI (2016) | 8% | 92% | -60.566 | $-2.211\pm22.343$ | 0.429 | 0.227 | $\times$ |
| E2ES-DR LAKSHMINARAYANAN ET AL. (2017) | 9% | 91% | -60.572 | $-1.499\pm21.500$ | 0.431 | 0.224 | $\times$ |
| DFB PEREIRA ET AL. (2020) | 4% | 39% | -18.114 | $-0.828\pm5.444$ | 0.513 | 0.125 | $\checkmark$ |
| BNET XIAO ET AL. (2023A) | 3% | 33% | -16.694 | $-4.882\pm4.817$ | 0.724 | 0.385 | $\checkmark$ |
| BNET-UP WANG ET AL. (2023B) | 2% | 35% | -23.252 | $-5.190\pm4.920$ | 0.726 | 0.532 | $\times$ |
| ABNET (OURS) | 0% | 100% | 1.455 | $6.132\pm2.181$ | 0.168 | 0.316 | $\checkmark$ |
| ABNET-ATT (OURS) | 0% | 100% | 4.198 | $8.053\pm1.449$ | 0.172 | 0.269 | $\checkmark$ |
| ABNET-SC (OURS) | 0% | 100% | 2.221 | $7.224\pm1.667$ | 0.130 | 0.256 | $\checkmark$ |

As shown in Table 3, the proposed ABNets can avoid crash onto obstacles with 100% obstacle passing rate, including the ABNet-sc that is trained in a scalable way with two ABNets (also shown by the scalable training in Fig. 5). This is because the ABNets can learn the correct steering control (the blue and green sine waves shown in Fig. 9 (right) in Appendix B.3) to avoid the obstacle without stopping in front of it. The DFB and BNet-related models learn a significant deceleration control (shown in Fig. 9) to avoid crashing onto obstacles, which explains why the corresponding obstacle passing rates are low compared to other models in Table 3 and why the blue trajectories (BNet) terminate near the obstacle in Fig. 5 (left). Nonetheless, there are still some crash cases in DFB and BNet models due to badly learned CBF parameters that make the inter-sampling effect (i.e., safety violation between discretized times) serious. Most importantly, our proposed ABNet can learn less

uncertain controls for this complicated task, as shown in Table 3, the scalable training in Fig. 5, and Fig. 9 (e.g., ABNet:blue or ABNet-att:green area v.s. BNet: red area). The ABNet-att can learn more consistent autonomous driving behavior than the ABNet due to the image attention setting, as shown by the magenta (ABNet-att) and cyan (ABNet) trajectories in Fig. 5 (left) and the green (ABNet-att) and blue (ABNet) areas in Fig. 9. **Ablation studies** on the robustness of our ABNets in terms of safety under high-noisy inputs (50% noise level) are given in Table 4 of Appendix B.3.

## 5 RELATED WORKS

**Scalability, merging and uncertainty in safe robot learning.** Machine learning has been widely used in robot control Bommasani et al. (2021) Singh et al. (2023) Wang et al. (2023a). Mixture of expert methods Shazeer et al. (2017) Riquelme et al. (2021) Zhou et al. (2022) are scalable but hard to retain the property (such as safety) of the models. The uncertainty resulting from noisy model input or dataset is preventing the deployment to real robots Loquercio et al. (2020) Kahn et al. (2017). To address this, predictive uncertainty quantification Gal & Ghahramani (2016) Lakshminarayanan et al. (2017), also a model merging approach, has been widely adopted. It has been shown to work well in vision-based autonomous driving under noisy input Wang et al. (2023b) using the Gaussian kernel with Scott's rule Scott (2015) to select bandwidth. The main challenge of this technique is that it may make the system lose performance guarantees, such as safety. Other model merging approaches Huang et al. (2023) Ramé et al. (2023) Wang et al. (2024) do not preserve safety either. We address the uncertainty and scalablibity problem using the proposed ABNets with provable safety.

**CBFs and set invariance.** In control theory, the set invariance has been widely adopted to prove and enforce the safety of dynamical systems (Blanchini, 1999) (Rakovic et al., 2005) (Ames et al., 2017) Xiao & Belta (2022) Xiao et al. (2023a). The Control Barrier Function (CBF) (Ames et al., 2017) Xiao & Belta (2022) is such a state of the art technique that can enforce set invariance (Aubin, 2009), (Prajna et al., 2007), (Wisniewski & Sloth, 2013), and transforms a nonlinear optimization problem to a quadratic problem that is very efficient to solve. CBFs originates from barrier functions that are originally used in optimization problems (Boyd & Vandenberghe, 2004). However, the CBF tends to make the system conservative (i.e., at the cost of performance) in order to enforce safety, and it is not scalable to build large models. Our proposed ABNet can address all these limitations.

**Safety in neural networks.** Safety is usually enforced using optimizations. Recently, differentiable optimizations show great potential for learning-based control with safety guarantees (Pereira et al., 2020; Amos et al., 2018; Xiao et al., 2023a; Liu et al., 2023). The quadratic program (QP) can be employed as a layer in the neural network, i.e., the OptNet (Amos & Kolter, 2017). The OptNet has been used with CBFs in neural networks as a safe filter controls (Pereira et al., 2020), in which CBFs themselves are not trainable, which can significantly limiting the learning capability. Neural network controllers with safety certificate have been learned through verification-in-the-loop training (Deshmukh et al., 2019; Zhao et al., 2021; Ferlez et al., 2020). However, this verification method cannot ensure to cover the whole state space. CBFs are also used in neural ODEs to equip them with specification guarantees Xiao et al. (2023b). None of these methods are scalable to larger models, and are subject to uncertainty, which the proposed ABNet can address.

## 6 CONCLUSIONS, LIMITATIONS AND FUTURE WORK

We propose a novel Attention BarrierNet (ABNet) that merge many safety-critical learning models while preserving the safety in this paper. The proposed ABNet is scalable to larger safe learning models, can achieve better performance, and is robust to input noise. We have demonstrated the effectiveness of the model on a series of robot control tasks. Nonetheless, our model still have a few limitations motivating for further research.

**Limitations.** First, all the ABNets have the same safety constraints. We will explore how to merge ABNets with different safety constraints in the future. Second, the ABNet also requires safety specifications that may be unknown in some robot control tasks, we may learn the safety specifications from data Robey et al. (2020), Srinivasan et al. (2020), and this can also be done in conjunction with ABNet. Finally, the model merging is done in the output space, future work will further focus on model merging with safety guarantees in the parameter space.

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

# A PROOF OF THEOREMS

**Theorem 3.1.** (**Safety of ABNets**) Given the multi-head BarrierNets formulated as in (6) s.t. (7). If the system (1) is initially safe (i.e., $b_j(x(t_0)) \geq 0, \forall j \in S$), then a control policy $u$ from the ABNet output (8) guarantees the safety of system (1), i.e., $b_j(x(t)) \geq 0, \forall j \in S, \forall t \geq t_0$.

**Proof:** The proof outline is to first show the existence of new HOCBF constraints (corresponding to all the safety specifications) that are defined over the output of the ABNet. Then, we can use Nagumo's theorem Nagumo (1942) to recursively show the forward invariance of each safety set in the HOCBFs, and this can eventually imply the satisfaction of the safety specifications $b_j(x) \geq 0, \forall j \in S$.

Since each control $u_k, k \in \{1, \ldots, h\}$ in the ABNet is obtained from solving the QP (6) s.t. (7), we have that the following constraint is satisfied:

$$L_f \psi_{j,m-1}(x,z|\theta_p) + [L_g \psi_{j,m-1}(x,z|\theta_p)]u_k + p_{m,k}(z_k|\theta_{p,k}^m)\alpha_{j,m}(\psi_{j,m-1}(x,z|\theta_p)) \geq 0, j \in S, \quad (10)$$

Multiplying the weight $w_k \geq 0$ to the last equation, we have

$$w_k L_f \psi_{j,m-1}(x,z|\theta_p) + w_k [L_g \psi_{j,m-1}(x,z|\theta_p)]u_k + w_k p_{m,k}(z_k|\theta_{p,k}^m)\alpha_{j,m}(\psi_{j,m-1}(x,z|\theta_p)) \geq 0, j \in S, \tag{11}$$

Taking a summation of the last equation over all $k \in \{1, \ldots, h\}$, the following equation establishes:

$$\sum_{k=1}^{h} w_k L_f \psi_{j,m-1}(x,z|\theta_p) + \sum_{k=1}^{h} w_k [L_g \psi_{j,m-1}(x,z|\theta_p)]u_k$$
$$+ \sum_{k=1}^{h} w_k p_{m,k}(z_k|\theta_{p,k}^m)\alpha_{j,m}(\psi_{j,m-1}(x,z|\theta_p)) \geq 0, j \in S, \tag{12}$$

Since $L_g \psi_{j,m-1}(x,z|\theta_p$ is a vector that is independent of $k$ and $\sum_{k=1}^{h} w_k = 1$, the last equation can be rewritten as:

$$L_f \psi_{j,m-1}(x,z|\theta_p) + L_g \psi_{j,m-1}(x,z|\theta_p) \left( \sum_{k=1}^{h} w_k u_k \right)$$
$$+ \sum_{k=1}^{h} w_k p_{m,k}(z_k|\theta_{p,k}^m)\alpha_{j,m}(\psi_{j,m-1}(x,z|\theta_p)) \geq 0, j \in S, \tag{13}$$

The summation of class $\mathscr{K}$ functions is also a class $\mathscr{K}$ function. Since $\alpha_{j,m}$ are class $\mathscr{K}$ functions, the $\sum_{k=1}^{h} w_k p_{m,k}(z_k|\theta_{p,k}^m)\alpha_{j,m}(\psi_{j,m-1}(x,z|\theta_p))$ is also a class $\mathscr{K}$ function over $\psi_{j,m-1}(x,z|\theta_p)$. Therefore, equations (13) are the **new HOCBF constraints** defined over the output of the ABNet, i.e., $\sum_{k=1}^{h} w_k u_k$. In other words, whenever $\psi_{j,m-1}(x,z|\theta_p) = 0$, we have

$$L_f \psi_{j,m-1}(x,z|\theta_p) + L_g \psi_{j,m-1}(x,z|\theta_p) \left( \sum_{k=1}^{h} w_k u_k \right) \geq 0, j \in S, \tag{14}$$

The controls (outputs of the ABNet) $\sum_{k=1}^{h} w_k u_k \equiv u$ are directly used to drive the system (1), and $z$ is taken as a piece-wise constant within discretized time intervals Xiao et al. (2023a). Therefore, the last equation can be rewritten as

$$\frac{\partial \psi_{j,m-1}(x,z|\theta_p)}{\partial x}(f(x) + g(x)u) = \frac{\partial \psi_{j,m-1}(x,z|\theta_p)}{\partial x}\dot{x} = \dot{\psi}_{j,m-1}(x,z|\theta_p) \geq 0, j \in S, \quad (15)$$

Since $b_j(x(t_0)) \geq 0$, we can always initialize the HOCBF definition such that $\dot{\psi}_{j,m-1}(x,z|\theta_p) \geq 0$ is satisfied at $t_0$ Xiao & Belta (2022). By Nagumo's theorem Nagumo (1942) and (13)-(15), we have that $\psi_{j,m-1}(x,z|\theta_p) \geq 0, \forall t \geq t_0$.

Recursively, we can show that $\psi_{j,i}(x,z|\theta_p) \geq 0, \forall t \geq t_0, \forall i \in \{0, \ldots, m-1\}$ from $i = m-1$ to $i = 0$. Since $b_j(x) = \psi_{j,0}(x,z|\theta_p)$ by (2), we have that $b_j(x(t)) \geq 0, \forall t \geq t_0, \forall j \in S$, which the safety guarantees of the ABNet for system (1). ∎

**Theorem 3.2.** (**Safety of merging of ABNets**) Given two ABNets with each formulated as in (8) and (6) s.t. (7), the merged model using the form as in (8) again guarantees the safety of system (1).

**Proof:** The proof outline is similar to that of Theorem 3.1. From each ABNet, we can show the existence of new HOCBF constraints (corresponding to all the safety specifications) that are defined over the output of each ABNet. Then we can again show the existence of another set of new HOCBF constraints (corresponding to all the safety specifications) that are defined over the output of the merged ABNet. Finally, we can also use Nagumo's theorem Nagumo (1942) to recursively show the forward invariance of each safety set in the HOCBFs, and this can eventually imply the satisfaction of the safety specifications $b_j(\boldsymbol{x}) \geq 0, \forall j \in S$.

The mathematical proof is similar to that of Theorem 3.1, and thus is omitted.

# B EXPERIMENT DETAILS

**Metrics used in all the tables Xiao et al. (2023b).** The SAFETY metric is defined as:

$$\text{SAFETY} = \min_k \{ \min_{t \in [t_0, T]} b(\boldsymbol{x}(t)) \}_k, k \in \{1, \dots, N\}, \quad (16)$$

where $N$ is the number of testing runs ($N = 100$ in this case). $T$ is the final time of each run. $b(\boldsymbol{x}) \geq 0$ is the safety constraint that is given explicitly in each experiment below.

The CONSER. metric is defined as

$$\text{CONSER. mean} = \text{mean}_k \{ \min_{t \in [t_0, T]} b(\boldsymbol{x}(t)) \}_k, k \in \{1, \dots, N\},$$

$$\text{CONSER. std} = \text{std}_k \{ \min_{t \in [t_0, T]} b(\boldsymbol{x}(t)) \}_k, k \in \{1, \dots, N\}. \quad (17)$$

The UNCERTAINTY metric for both controls are calculated by:

$$u_i \text{ UNCERTAINTY} = \text{mean}_{t \in [t_0, T]} \{ \text{std}_k \{ u_i(t) \}_k, k \in \{1, \dots, N\} \}, i \in \{1, 2\}. \quad (18)$$

All the class $\mathscr{K}$ functions in the BarrierNets/ABNets are implemented as linear functions with trainable slopes.

## B.1 2D ROBOT OBSTACLE AVOIDANCE

**Models.** All the models include fully connected layers of shape [5, 128, 32, 32, 2] with RELU as activation functions. There are some additional layers of differentiable QPs in other models (other than E2E-related models). The model input is the system state and the goal.

**Training and Dataset.** The dataset includes 100 trajectories, and each trajectory has 137 trajectory points. The ground truth controls (i.e., training labels) are obtained via solving HOCBF-based QPs Xiao & Belta (2022). We use *Adam* as the optimizer to train the model with a MSE loss function and a learning rate 0.001. We use the *QPFunction* from the OptNet Amos & Kolter (2017) to solve the dQPs. The training time of the ABNet is about 1 hour for 20 epochs on a RTX-3090 computer.

**Robot dynamics and safety constraints.** We employ the bicycle model as the robot dynamics:

$$\underbrace{\begin{bmatrix} \dot{x}(t) \\ \dot{y}(t) \\ \dot{\theta}(t) \\ \dot{v}(t) \end{bmatrix}}_{\dot{\boldsymbol{x}}(t)} = \underbrace{\begin{bmatrix} v(t)\cos\theta(t) \\ v(t)\sin\theta(t) \\ 0 \\ 0 \end{bmatrix}}_{f(\boldsymbol{x})} + \underbrace{\begin{bmatrix} 0 & 0 \\ 0 & 0 \\ 1 & 0 \\ 0 & 1 \end{bmatrix}}_{g(\boldsymbol{x})} \underbrace{\begin{bmatrix} u_1(t) \\ u_2(t) \end{bmatrix}}_{\boldsymbol{u}} \quad (19)$$

where $(x, y) \in \mathbb{R}^2$ denotes the 2D location of the robot, $\theta \in \mathbb{R}$ is the heading angle of the robot, $v \in \mathbb{R}$ is the linear speed of the robot. $u_1, u_2$ are the angular speed and acceleration controls, respectively.

The safety constraint of the robot is defined as:

$$b(\boldsymbol{x}) = (x - x_0)^2 + (y - y_0)^2 - R^2 \geq 0, \quad (20)$$

where $(x_0, y_0) \in \mathbb{R}^2$ is the 2D location of the obstacle, and $R > 0$ is its size.

**Acceleration control profiles.** We show the acceleration control profiles in Fig. 6. The corresponding uncertainty is also significantly decreased with the proposed ABNet.

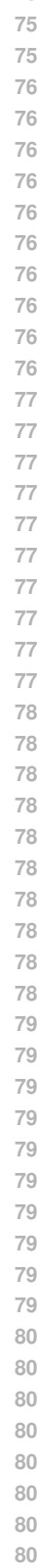

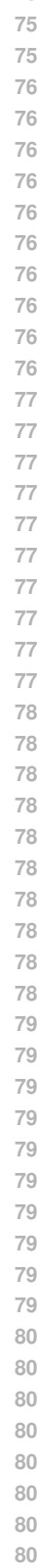

Figure 6: 2D robot obstacle avoidance acceleration control profiles and their distributions. The controls are subject to input noise, and thus are non-smooth. All the testings are done in a closed-loop fashion, i.e., the model outputs are directly used to control the robot.

## B.2 SAFE ROBOT MANIPULATION

**Models.** All the models include fully connected layers of shape [6, 128, 256, 128, 128, 32, 32, 2] with RELU as activation functions. There are some additional layers of differentiable QPs in other models (other than E2E-related models). The model input is the system state and the goal.

**Training and Dataset.** The dataset includes 1000 trajectories, and each trajectory has about 350 trajectory points. The ground truth controls (i.e., training labels) are obtained via solving HOCBF-based QPs Xiao & Belta (2022). We use *Adam* as the optimizer to train the model with a MSE loss function and a learning rate 0.001. We use the *QPFunction* from the OptNet Amos & Kolter (2017) to solve the dQPs. The training time of the ABNet is about 2 hours for 10 epochs on a RTX-3090 computer.

**Robot dynamics and safety constraints.** We employ the following model as the manipulator dynamics:

$$
\underbrace{\begin{bmatrix} \dot{\theta}_1 \\ \dot{\omega}_1 \\ \dot{\theta}_2 \\ \dot{\omega}_2 \end{bmatrix}}_{\dot{\boldsymbol{x}}} = \underbrace{\begin{bmatrix} \omega_1 \\ 0 \\ \omega_2 \\ 0 \end{bmatrix}}_{f(\boldsymbol{x})} + \underbrace{\begin{bmatrix} 0 & 0 \\ 1 & 0 \\ 0 & 0 \\ 0 & 1 \end{bmatrix}}_{g(\boldsymbol{x})} \underbrace{\begin{bmatrix} u_1 \\ u_2 \end{bmatrix}}_{\boldsymbol{u}} \tag{21}
$$

where $(\theta_1, \theta_2) \in \mathbb{R}^2$ denotes the angles of the two-link manipulator joints, $(\omega_1, \omega_2) \in \mathbb{R}^2$ is the angular speed of the two-link manipulator joints, $u_1, u_2$ are the angular acceleration controls corresponding to the two joints, respectively.

The safety constraint of the robot is defined as:

$$
b(\boldsymbol{x}) = (l_1 \cos \theta_1 + l_2 \cos \theta_2 - x_0)^2 + (l_1 \sin \theta_1 + l_2 \sin \theta_2 - y_0)^2 - R^2 \ge 0, \tag{22}
$$

where $(x_0, y_0) \in \mathbb{R}^2$ is the location of the obstacle, and $R > 0$ is its size. $l_1 > 0, l_2 > 0$ are the length of the two links of the manipulator, respectively. In the current setting, the non-collision of the end-effector implies the non-collision of the link. Therefore, we only need to consider the safety of the end-effector. We show both the $u_1, u_2$ control profiles in Fig. 7 to demonstrate the advantage of the proposed ABNet. The metric definitions are the same as in the 2D robot obstacle avoidance, and the number of testing runs is $N = 100$.

## B.3 VISION-BASED END-TO-END AUTONOMOUS DRIVING

**Models.** All the models include CNN ([[3, 24, 5, 2, 2], [24, 36, 5, 2, 2], [36, 48, 3, 2, 1], [48, 64, 3, 1, 1], [64, 64, 3, 1, 1]]) and LSTM layers (size: 64) and some fully connected layers of shape [32,

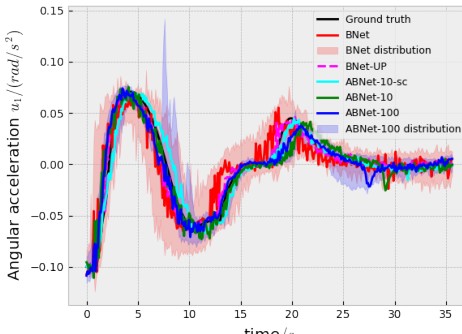 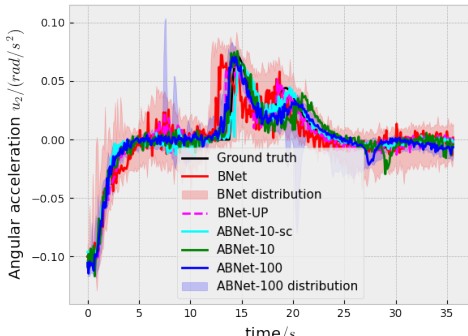

Figure 7: Robot manipulation joint control profiles and their distributions. The controls are subject to input noise, and thus are non-smooth. All the testings are done in a closed-loop fashion, i.e., the model outputs are directly used to control the manipulator.

32, 2] ×2 with RELU as activation functions. The dropout rates for both CNN and fully connected layers are 0.3. There are some additional layers of differentiable QPs in other models (other than E2E-related models). The model input is the front-view RGB images (shape: $3 \times 45 \times 155$) of the ego vehicle, and the outputs are the steering rate and acceleration controls of the vehicle.

**Training and Dataset.** The dataset is open-sourced including 0.4 million image-control pairs from a closed-road sim-to-real driving field. Static and parked cars of different types and colors are used as obstacles in the dataset. The dataset is collected from the VISTA simulator Amini et al. (2022). The ground truth controls (i.e., training labels) are obtained via solving a nonlinear model predictive control (NMPC). We use *Adam* as the optimizer to train the model with a MSE loss function and a learning rate 0.001. We use the *QPFunction* from the OptNet Amos & Kolter (2017) to solve the dQPs. The training time of the ABNet is about 15 hours for 5 epochs on a RTX-3090 computer.

**Brief introduction to VISTA.** VISTA is a sim-to-real driving simulator that can generate driving scenarios from real driving data Amini et al. (2022). The VISTA allows us to train our model with guided policy learning. This learning method has been shown to work for model transfer to a full-scale real autonomous vehicle. There three steps to generate the data: (i) In VISTA, we randomly initialize the locations and poses of ego- and ado-cars that are associated with the real driving data; (ii) we use NMPC to collect ground-truth controls (training labels) with corresponding states, and (iii) we collect front-view RGB images along the trajectories generated from NMPC.

**Vehicle dynamics and safety constraints.** The vehicle dynamics are specified with respect to a reference trajectory Rucco et al. (2015), such as the lane center line. The two most important states are the along-trajectory progress $s \in \mathbb{R}$ and the lateral offset distance $d \in \mathbb{R}$ of the vehicle center with respect to the trajectory. The dynamics are defined as:

$$\underbrace{\begin{bmatrix} \dot{s} \\ \dot{d} \\ \dot{\mu} \\ \dot{v} \\ \dot{\delta} \end{bmatrix}}_{\dot{\boldsymbol{x}}} = \underbrace{\begin{bmatrix} \frac{v\cos(\mu+\beta)}{1-d\kappa} \\ v\sin(\mu+\beta) \\ \frac{v}{l_r}\sin\beta - \kappa\frac{v\cos(\mu+\beta)}{1-d\kappa} \\ 0 \\ 0 \end{bmatrix}}_{f(\boldsymbol{x})} + \underbrace{\begin{bmatrix} 0 & 0 \\ 0 & 0 \\ 0 & 0 \\ 1 & 0 \\ 0 & 1 \end{bmatrix}}_{g(\boldsymbol{x})} \underbrace{\begin{bmatrix} u_1 \\ u_2 \end{bmatrix}}_{\boldsymbol{u}}, \tag{23}$$

where $\mu$ is the local heading error of the vehicle with respect to the reference trajectory, $v$ is the linear speed of the vehicle, $\kappa$ is the curvature of the trajectory at the progess $s$. $l_r$ is the length of the vehicle from the tail to the center, $\beta = \arctan\left(\frac{l_r}{l_r+l_f}\tan\delta\right)$, where $l_f$ is the length of the vehicle from the head to the center. $u_1, u_2$ are the steering rate and acceleration controls of the vehicle, respectively.

The safety constraint of the vehicle is defined as:

$$b(\boldsymbol{x}) = (s-s_0)^2 + (d-d_0)^2 - R^2 \geq 0, \tag{24}$$

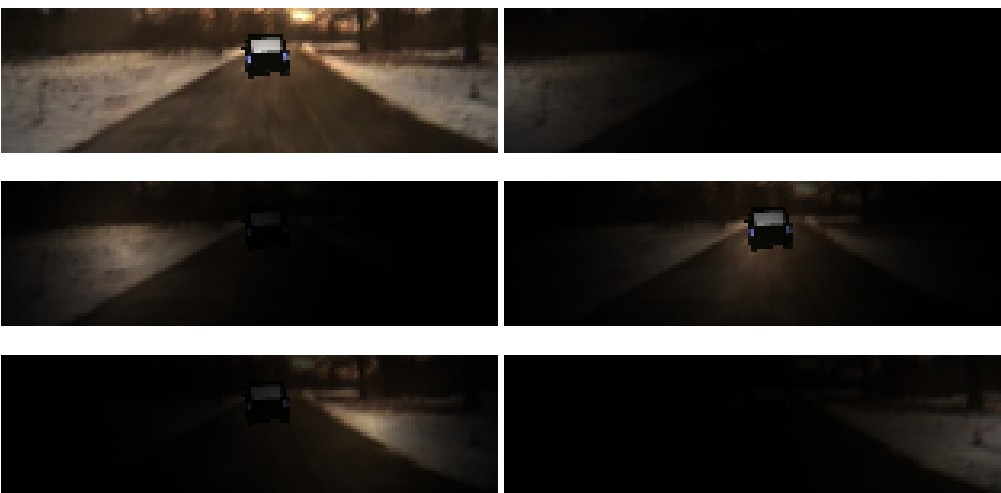

Figure 8: Attention-based image observations for the ABNet-att model. From left to right and top to down: attentions on full image, left-most part, left lane boundary, lane center, right lane boundary, and right-most part.

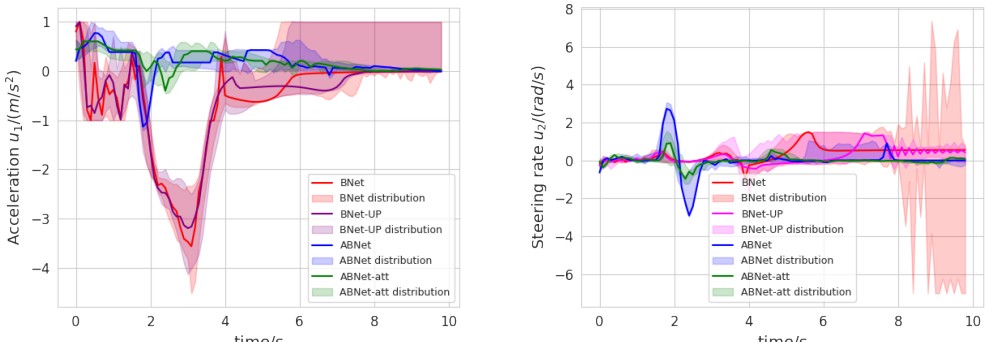

Figure 9: Vision-based end-to-end autonomous driving closed-loop testing control profiles. The models directly take images as inputs, and output controls for the vehicle. All the testings are done in closed-loop in VISTA.

where $(s_0, d_0) \in \mathbb{R}^2$ is the location of the obstacle in the curvi-linear frame (i.e., defined with respect to the reference trajectory), and $R > 0$ defines its size that is chosen such that the satisfaction of the above constraint can make the ego vehicle avoid crashing onto the obstacle.

**Closed-loop testing.** We test all of our models in a closed-loop manner in VISTA. In other words, at each time step, we get the front-view RGB image observation from VISTA. Then, the model generates a control based on the image. Finally, the control is used to drive the "virtual" vehicle in VISTA. This process is done recursively until the final time. The total number of testing runs is $N = 100$ for all the tables. The obstacles are randomly initialized (in uniform probability distribution) with lateral distance $d_0$ ranges from $\pm 0.1m$ to $\pm 1.5m$. In Figs. 5 and 9, the ego vehicle is randomly initialized with $d \in [-0.5, 0.5]m$ (in uniform probability distribution).

**Image observations for the ABNet-att model.** We generate the attention-based observations as shown in Fig. 8. Each of the attention images may play an important role in a specific driving scenario (e.g., attention on the left-most part may be crucial for sharp-left turn).

**Acceleration control profiles.** We present both the acceleration control and steering rate control profiles in Fig. 9. Both the BNet and BNet-UP models have forced the ego vehicle to have a large deceleration instead of making it to pass the obstacle using the steering control when the vehicle approaches the obstacle. This can make the ego vehicle get stuck at the obstacles, and thus, the obstacle passing rate (as shown in Table 3) is low in these two models.

Table 4: Ablation study: vision-based end-to-end autonomous driving closed-loop testing **under noise** and comparisons with benchmarks. Items in the first row are short for obstacle crash rate (CRASH), Obstacle passing rate (PASS), satisfaction of safety constraints where non-negative values mean safety guarantees (SAFETY), system conservativeness (CONSER.), acceleration control $u_1$ uncertainty ($u_1$ UNCERTAINTY), steering rate control $u_2$ uncertainty ($u_2$ UNCERTAINTY), and theoretical safety guarantees (THEORET. GUAR.) respectively. In the model column, items are short for single vanilla end-to-end driving model (V-E2E), E2Es merged with Monte-Carlo Dropout (E2Es-MCD), E2Es merged with deep resembles (E2Es-MERG), deep forward and backward model (DFB), single BarrierNet (BNET), BarrierNet policies with uncertainty propagation (BNET-UP), ABNet with 10 heads (ABNET), ABNet with attention images and 10 heads (ABNET-ATT), ABNET-SC denotes our ABNet first trained with ABNET-ATT scaled by ABNET (20 heads)respectively. The safety metric is defined as the **minimum** value of the safety specification $b_j(\boldsymbol{x}), j \in S$ among all runs. The conservativeness metric is defined as the **mean** (with std) of the minimum value (in each run) of the safety specification $b_j(\boldsymbol{x}), j \in S$ among all runs. The uncertainty metrics for both $u_1$ and $u_2$ are measured by the standard deviations of the model outputs (two controls) among all runs.

| MODEL | CRASH ($\downarrow$) | PASS ($\uparrow$) | SAFETY ($\geq 0$) | CONSER. ($\geq 0$ & $\downarrow$) | $u_1$ UNCERTAINTY ($\downarrow$) | $u_2$ UNCERTAINTY ($\downarrow$) | THEORET. GUAR. |
|---|---|---|---|---|---|---|---|
| V-E2E AMINI ET AL. (2022) | 31% | 69% | -59.455 | $-8.932\pm19.741$ | 0.529 | 0.239 | $\times$ |
| E2Es-MCD GAL & GHAHRAMANI (2016) | 28% | 72% | -58.405 | $-8.116\pm20.802$ | 0.524 | 0.232 | $\times$ |
| E2Es-DR LAKSHMINARAYANAN ET AL. (2017) | 27% | 73% | -60.267 | $-8.781\pm20.910$ | 0.512 | 0.225 | $\times$ |
| DFB PEREIRA ET AL. (2020) | 1% | 37% | -13.281 | $-0.256\pm4.348$ | 0.482 | 0.127 | $\checkmark$ |
| BNET XIAO ET AL. (2023A) | 23% | 37% | -45.415 | $-9.114\pm13.382$ | 0.730 | 0.316 | $\checkmark$ |
| BNET-UP WANG ET AL. (2023B) | 24% | 39% | -44.634 | $-8.866\pm13.167$ | 0.747 | 0.278 | $\times$ |
| ABNET (OURS) | 0% | 100% | 4.268 | $8.315\pm2.147$ | 0.151 | 0.326 | $\checkmark$ |
| ABNET-ATT (OURS) | 0% | 100% | 5.986 | $7.032\pm0.405$ | 0.118 | 0.213 | $\checkmark$ |
| ABNET-SC (OURS) | 0% | 100% | 4.118 | $7.515\pm1.120$ | 0.128 | 0.255 | $\checkmark$ |

**Ablation studies on the model robustness in terms of safety under noisy input.** To further test the model safety robustness, we add random noise (50% magnitude of the image values) to all the image observations. The results are presented in Table 4. Our proposed ABNets can still guarantee the safety of the vehicle under noisy input (0% crash rate), while the crash rates using other models significantly increase except the DFB model. This is because the HOCBFs in the DFB model are not trainable, and the corresponding parameters are fixed. Badly trained HOCBFs could make the method fail to guarantee safety due to the inter-sampling effect.

