# OpenReview forum: "ABNet: Attention BarrierNet for Safe and Scalable Robot Learning"
_ICLR.cc/2025/Conference — Submitted to ICLR 2025_

### Official Review · Reviewer_3NQu · 2024-10-31

**Soundness:** 2
**Presentation:** 2
**Contribution:** 2
**Rating:** 3
**Confidence:** 3

**Summary:**

The paper introduces ABNet (Attention BarrierNet), a novel neural network architecture designed for safe and scalable robot learning.
- ABNet combines multiple "BarrierNet" heads that can each learn safe control policies while focusing on different aspects of the input observation
- The architecture allows for merging multiple safety-critical models while preserving safety guarantees
- It's scalable and can be trained incrementally to build larger foundational safe models

**Strengths:**

1. Provable theoretical safety guarantee using forward invariance.

2. Better safety performance across multiple benchmarks.

**Weaknesses:**

Most of the weaknesses are well explained in the limitation section. In addition to that:

1. It is not well explained how attention is used in the ABNet. See question 1.

2. It is uncertain how effective this self-attention is rather than the contribution of scaling model size. See question 2.

3. The control affine assumption seems strong. The approach heavily relies on the assumption, because it linearly interpolates the controls.

**Questions:**

1. About the definition of attention

1a. How is the self-attention used for ABNet? Usually, self-attention refers to the one in the transformer. I think, in general, ABNet is not restricted to being a transformer. So how was it used for the ABNet? Is it used for both state-based task and vision-based task?

1b. A follow-up question: Is Fusion also part of the attention? If so, I think it is not the usual self-attention. Normally, self-attention calculates the score based on the softmax between K and Q. In this paper, it is simply a weighting coefficient. The fusion looks more similar to model ensemble rather than self-attention.

2. About the effectiveness of attention

2a. I wonder whether the approach's effectiveness comes mainly from enlarging model size instead of using this attention mechanism. Can we have some experiments that enlarge the size of the BNet to the same size as those best models, e.g., ABNet-10 or ABNet-100, and compare this large BNet to the same-size ABNet, w.r.t. the existing benchmarks? It would be appreciated if the model parameter size for each method could be a column in the table.

3. About the experiment setting

3a. Is imitation learning used for the experiment? If so, do we assume a safe enough reference controller should be known?

3b. The vision-based task seems the most interesting one, and I wish to know more about it. If I understand correctly, the x space here is the image space. How are the dynamics defined (or are they even unknown), and are they control-affine?

3c. I notice in the setting, "with uniformly distributed noise, 10% of the input magnitude in testing". Why are they needed for the testing? Is it for the claim of robustness of the control? I wonder what the performance will be if we simply make the training and testing dynamics the same.

4. About theoretical guarantee

4a. A general problem with the learning-based CBF is that they are guaranteed to be a CBF only when they are trained very well to satisfy the forward invariance property. If out-of-distribution states happen, or the neural CBF fails to fit on the training dataset, then it is not a theoretical CBF anymore. Just as a discussion, does this work circumvent this problem? Meanwhile, I don't think this is a big issue since almost all neural CBFs have this problem.

---

### Official Review · Reviewer_xzNy · 2024-11-01

**Soundness:** 2
**Presentation:** 2
**Contribution:** 2
**Rating:** 5
**Confidence:** 4

**Summary:**

This paper addresses two critical aspects of robot learning: safety and scalability. While traditional barrier-based models have been utilized to ensure safety in robotic systems, they present the following challenges: lack of scalability, learning stability, slow training speed and sensitivity to noise. To overcome these issues, the paper proposes the Attention BarrierNet (ABNet). This model guarantees the necessary safety for robotic control while being scalable and capable of incremental learning. ABNet offers a novel approach that addresses the limitations of existing barrier-based models by balancing safety and performance. Experimental results demonstrate the superiority of ABNet across various tasks, particularly highlighting its ability to maintain robust control signals even in noisy environments.

**Strengths:**

1.  Scalability: ABNet utilizes multiple heads, allowing for the incremental construction of large-scale models suitable for complex tasks.
2.  Safety Assurance: The outputs from each head are combined in a way that ensures mathematical safety guarantees.
3.  Robustness to Noise: ABNet provides stable outputs even with noisy input data, as evidenced by smooth signals in testing.
4.  Parallel Learning: Each head can be trained independently, maximizing learning efficiency.

**Weaknesses:**

1.  Uniform Safety Constraints: All heads currently operate under the same safety constraints, limiting the model's ability to incorporate diverse constraint types.
2.  Uncertainty in Safety Specifications: In certain robotic control tasks, safety specifications are not clearly defined, indicating the need for further research.
3.  Output Space Combination: The model currently combines outputs only in the output space. Further research is required to explore safe combinations within the parameter space.

**Questions:**

## Theoretical Concerns

1.  Linear Combination of Policy Outputs

    ABNet generates its final control output by linearly combining the outputs from multiple heads. However, there are concerns about whether such a linear combination is effective for complex nonlinear problems. Many real-world environments involve significant nonlinearity and complex interactions, which may not be fully captured by linear combinations.

2.  Generalization of Safety Guarantees

    While the paper provides theoretical guarantees that the combined outputs ensure safety, there is insufficient explanation regarding whether these guarantees hold in novel situations not covered by the training data. If the safety constraints do not generalize across all scenarios, the model may malfunction in unexpected environments.

3.  Gap Between Mathematical Models and Reality

    ABNet relies on mathematical optimization (specifically Quadratic Programming, or QP) to implement safety control. However, real-world uncertainties, noise, and sensor errors may reduce the model’s performance. There is a concern about whether the mathematical models, which do not account for such uncertainties, can achieve similar performance in practice.


----------

## Experimental Concerns

1.  Gap Between Simulation and Real-World Environments

    Most of the experiments in the paper were conducted in simulated environments. However, there are significant differences between simulations and real-world conditions. For example, sensor errors and unexpected scenarios, which may not be fully replicated in simulations, could affect safety in practice.

2.  Interpretation of Performance Metrics

    The paper presents metrics on rewards and safety but may have overlooked potential trade-offs between safety and performance in specific tasks. For instance, to achieve high rewards, the model might subtly violate some safety constraints, but this trade-off has not been analyzed in detail.

3.  Limitations of Comparative Experiments

    While ABNet's performance was compared with several existing models, there is a possibility that the comparison was not exhaustive. The inclusion of more recent reinforcement learning or control models would enhance the reliability of the results.

4.  Overfitting Risks

    Since ABNet trains each head individually and combines them linearly, there is a potential risk of overfitting to the training data. This could limit the model’s generalization to unseen scenarios, leading to performance degradation in untested environments.


----------

## Comprehensive Concerns

-   Generalization and Data Limitations: If the training data and safety constraints are limited, the model might struggle to handle the wide range of situations encountered in real-world applications.
-   Real-Time Control Challenges: The QP-based optimization required for safety control may not be suitable for real-time control due to computational delays. The paper lacks a detailed discussion of latency issues, which are critical for real-time robotic control.
-   Probabilistic Safety Guarantees: If safety is guaranteed only probabilistically or as an expected value, rare but critical safety violations could occur. The paper does not adequately address how to handle these rare but potentially catastrophic violations.

These theoretical and experimental concerns are crucial for evaluating the reliability of ABNet’s proposed methodology and results. Future research should focus on conducting experiments in more diverse environments and incorporating complex real-world scenarios. Additionally, further studies are needed to ensure real-time performance and provide clearer safety guarantees.

---

### Official Review · Reviewer_GSb4 · 2024-11-04

**Soundness:** 3
**Presentation:** 3
**Contribution:** 2
**Rating:** 3
**Confidence:** 3

**Summary:**

The paper introduces ABNet, a novel framework that leverages attention mechanisms to address varying input patterns while integrating barrier functions to ensure the system state remains within a safety set, achieving forward invariance. The proposed method aims to enhance scalability and robustness in robot learning by allowing each head of BarrierNet to focus on different aspects of the observation space, potentially enabling safe control policies in diverse environments.

**Strengths:**

1. The approach explicitly incorporates barrier functions into neural network training, ensuring safety constraints are satisfied.
2. The modular architecture of ABNet with multiple attention heads allows scalable, incremental learning, which is promising for building complex, safe models in stages.
3. The method demonstrates robustness to noise, yielding lower variance in performance.

**Weaknesses:**

1. **Lack of Optimality Guarantees**: The method does not appear to ensure optimal task performance. As stated in the paper, "we use NMPC to collect ground-truth controls (training labels) with corresponding states," implying that the upper limit of ABNet's task performance is constrained by the performance of NMPC (e.g., minimum time to reach a target). Additionally, optimality does not seem to be the primary focus in training. By employing imitation learning with barrier functions, safety appears to be prioritized, which could further impact task performance. Is there a mechanism to balance task performance and safety? Furthermore, the criteria for selecting the penalty functions $\alpha$ in Equation 7 are not well explained. How are these values chosen, and how does the method balance conservatism and performance optimality?

2. **Simplicity of Scenarios Considered**: The experimental scenarios are relatively simplistic, involving static environments with limited dynamics, and lack sufficient qualitative analysis, such as video demonstrations comparing the method with baselines. Additionally, Figure 5 does not clearly illustrate the performance differences between ABNet and methods like MPC or BNet. Moreover, while Figure 5 includes results from MPC, Table 3 does not provide a corresponding quantitative comparison. For the method to be applicable to more complex, real-world scenarios—such as autonomous driving with dynamic obstacles like vehicles or pedestrians—consideration of the environment's external dynamics is crucial. This raises concerns about the scalability of ABNet. The authors could consider testing in more sophisticated simulators, such as [1-2], to better demonstrate the method's adaptability and robustness in dynamic environments.

3. **Contribution of Attention Mechanism**: The primary contribution appears to be the integration of attention into BarrierNet. The authors are encouraged to provide a detailed explanation of why this integration is non-trivial. Replacing BarrierNet’s structure with an attention mechanism might seem straightforward, as it could be perceived as a simple network architecture change. Were there specific challenges or technical hurdles encountered when integrating attention into BarrierNet? Did any particular techniques make this integration effective, ? Discussing these aspects would clarify the unique benefits of the proposed approach.

[1] CARLA: An open urban driving simulator

[2] Nuscenes: A multimodal dataset for autonomous driving

**Questions:**

1.  Would the method still be effective in environments that require sophisticated contact handling? How might this approach be extended to humanoid robots, especially in scenarios involving complex external interactions, such as full-body dynamics with external force feedback?

2. To my knowledge, large transformer models are increasingly applied in autonomous driving scenarios involving dynamic external agents, as referenced in [3]. This approach uses transformers as a backbone for safety planning based on sampled trajectories, ensuring collision-free paths while closely resembling NMPC planners or human driving behaviors. How do the authors view the online planning, and could barrier functions be integrated to enhance safety?

3. The method’s applicability appears limited by the requirement for a known, differentiable constraint function. In many real-world scenarios, defining such a function can be highly complex or may even resemble a black-box system. For example, ensuring that large language models avoid generating outputs contrary to human intentions necessitates defining a "human-safe" constraint function, which is inherently difficult.

[3] Planning-oriented Autonomous Driving

---

### Official Review · Reviewer_Ux7C · 2024-11-05

**Soundness:** 3
**Presentation:** 3
**Contribution:** 3
**Rating:** 6
**Confidence:** 3

**Summary:**

This paper proposes a model for safe and scalable robot learning. It addresses the limitations of traditional barrier-based methods, which are not scalable and may produce unstable outputs under noisy inputs. The proposed ABNet incorporates multiple "heads," each focusing on different aspects of the input to generate control policies safely and efficiently. This architecture enables incremental, scalable training without sacrificing safety guarantees. The paper presents theoretical proofs of ABNet’s safety and demonstrates its effectiveness across various robotic tasks, including 2D obstacle avoidance, safe manipulation, and vision-based autonomous driving, showing improved performance and robustness compared to existing methods.

**Strengths:**

1. The paper rigorously evaluates ABNet across multiple domains, using well-defined metrics and benchmarks to validate its performance and safety claims.

2. The integration of attention mechanisms into barrier-based models for robot learning is novel. By focusing each "head" on specific aspects of the input, ABNet allows for scalable, efficient training without needing to construct a large model from scratch, a unique approach in safe robot learning.

3. ABNet's scalability and robustness to noise make it highly valuable for real-world robot applications, particularly in safety-critical scenarios like autonomous driving.

**Weaknesses:**

1. The mathematical explanations, especially regarding HOCBFs and the attention mechanisms, are dense and may be challenging for some readers to follow. More intuitive explanations or diagrams could be better.

2. All heads in the ABNet share the same safety constraints. Expanding the model to allow for varying constraints in different heads could make it even more flexible and applicable to a wider array of tasks.

3. While ABNet’s scalability is demonstrated in 2D tasks and a simulated autonomous driving environment, it would benefit from testing in real-world environments to further validate its robustness and generalizability.

**Questions:**

1. How does ABNet handle situations where the safety constraints of different heads conflict? For example, if one head focuses on obstacle avoidance and another on goal attainment, how are these constraints balanced?

2. Could you provide further clarification on the choice of penalty functions in the HOCBFs? Specifically, how sensitive is the model's performance to different choices of these functions?

3. Has any analysis been done on the computational cost of scaling up ABNet to a large number of heads, especially in real-time applications like autonomous driving?

---

### Meta-Review · Area_Chair_9N1W · 2024-12-19

**Metareview:**

This paper proposed a 'safe learning' approach to learning robot control policies. The reviewers found a variety of issues with the paper, and the authors did not respond to these issues. In light of that, and the fact that no reviewer strongly advocated acceptance, I recommend that the paper be rejected in its current form (and that the authors try to address the concerns of the reviewers to improve the paper for a potential resubmission).

**Additional Comments On Reviewer Discussion:**

No author responses were provided, therefore no reviewer discussion occurred.

---

### Decision · Program_Chairs · 2025-01-22

Reject